# Reproducibility Study Of Learning Fair Graph Representations Via Automated Data Augmentations

**Thijmen Nijdam**                                                  *thijmen.nijdam@student.uva.nl*
*University of Amsterdam*

**Juell Sprott**                                                       *juell.sprott@student.uva.nl*
*University of Amsterdam*

**Taiki Papandreou-Lazos**                              *taiki.papandreou-lazos@student.uva.nl*
*University of Amsterdam*

**Jurgen de Heus**                                               *jurgen.de.heus@student.uva.nl*
*University of Amsterdam*

*Reviewed on OpenReview: https://openreview.net/forum?id=4WiqHopXQX*

## Abstract

In this study, we undertake a reproducibility analysis of "Learning Fair Graph Representations Via Automated Data Augmentations" by Ling et al. (2022). We assess the validity of the original claims focused on node classification tasks and explore the performance of the Graphair framework in link prediction tasks. Our investigation reveals that we can partially reproduce one of the original three claims and fully substantiate the other two. Additionally, we broaden the application of Graphair from node classification to link prediction across various datasets. Our findings indicate that, while Graphair demonstrates a comparable fairness-accuracy trade-off to baseline models for mixed dyadic-level fairness, it has a superior trade-off for subgroup dyadic-level fairness. These findings underscore Graphair's potential for wider adoption in graph-based learning. Our code base can be found on GitHub at https://github.com/juellsprott/graphair-reproducibility.

## 1 Introduction

Graph Neural Networks (GNNs) have become increasingly popular for their exceptional performance in various applications (Hamaguchi et al., 2017; Liu et al., 2022; Han et al., 2022a). A key application area is graph representation learning (Grover & Leskovec, 2016; Hamilton, 2020; Han et al., 2022b), where a significant concern is the potential for GNNs to inherit or amplify biases present in the graph representation training data. This can lead to discriminatory model behavior (Dai & Wang, 2022). To address this issue, Ling et al. (2022) introduced Graphair, an automated data augmentation technique aimed at learning fair graph representations without maintaining biases from the training data.

This work evaluates the main claims made by Ling et al. (2022), which were based solely on the performance of the framework in node classification tasks. We expand our evaluation by conducting additional experiments to assess the adaptability and generalizability of Graphair through its application to a different downstream task, namely, link prediction. This approach allows us to further test the performance and fairness of the embeddings.

For this purpose, we apply Graphair to new real-world datasets, adapting certain aspects of the framework and fairness metrics to suit link prediction. Our contributions are summarized as follows:

- We replicate the original experiments to assess the reproducibility of the primary claims. We find that one of the three claims made by the original authors was partially reproducible, while the remaining claims are fully verified.

- We adapt the Graphair framework for link prediction on various real-world datasets, which required modifications to both the framework and the fairness metrics. These adjustments provided valuable insights into Graphair's adaptability and generalizability to another downstream task. Our findings suggest that Graphair achieves a superior trade-off in one of the fairness metrics used for this task.

## 2 Scope of reproducibility

The original paper *Learning Fair Graph Representations Via Automated Data Augmentations* by Ling et al. (2022) introduces Graphair, an innovative automated graph augmentation method for fair graph representation learning. This approach stands out from prior methods (Luo et al., 2022; Zhao et al., 2022; Agarwal et al., 2021) by utilizing a dynamic and automated model to generate a new, fairer version of the original graph, aiming to balance fairness and informativeness (Ling et al., 2022).

In this work, we study the reproducibility of this paper. Besides examining the three main claims, we also assess the adaptability and effectiveness of Graphair in a different context, namely link prediction. This extension tests the framework's ability to maintain fairness and informativeness in a different downstream tasks. We will test this on a variety of datasets. The claims and our extension are as follows:

- Claim 1: Graphair consistently outperforms state-of-the-art baselines in node classification tasks for real-world graph datasets. Our extension evaluates whether this superior performance extends to link prediction tasks as well.

- Claim 2: Both fair node features and graph topology structures contribute to mitigating prediction bias.

- Claim 3: Graphair can automatically generate new graphs with fair node topology and features.

## 3 Methodology

### 3.1 Graphair

The *Graphair* framework introduces a novel approach to graph augmentation, focusing on the dual objectives of maintaining information richness and ensuring fairness. It incorporates a model $g$, which adeptly transforms an input graph $G = \{A, X, S\}$, where $A$ represents the adjacency matrix, $X$ the node features, and $S$ the sensitive attributes, into an augmented counterpart $G' = \{A', X', S\}$. This transformation utilizes two main operations: $T_A$ for adjusting the adjacency matrix $A$ and $T_X$ for masking node features $X$. A GNN-based encoder $g_{enc}$ precedes these transformations, tasked with extracting deep embeddings to inform these operations. The resulting augmented graph $G'$ is designed to capture the core structural and feature-based elements of the original graph $G$, while simultaneously keeping fairness principles to prevent the propagation of sensitive attribute information. This objective is achieved by training an adversarial model and using contrastive loss to optimize $G'$. An overview of the Graphair framework is presented in Figure 5 in Appendix A.1.

#### 3.1.1 Adversarial Training for Fairness

To ensure the fairness of the augmented graph, the model employs an adversarial training strategy. The adversarial model $k$ learns to predict sensitive attributes $S$ from the graph's features. Simultaneously, the representation encoder $f$ and augmentation model $g$ are optimized to minimize the predictive capability of the adversarial model, effectively removing biases from the augmented graph. The primary goal of adversarial training is to guide the encoder in generating representations that are free from sensitive attribute information. This optimization can be formally defined as follows:

$$\min_{g,f} \max_k L_{\text{adv}} = \min_{g,f} \max_k \frac{1}{n} \sum_{i=1}^{n} \left[ S_i \log \hat{S}_i + (1 - S_i) \log(1 - \hat{S}_i) \right] \tag{1}$$

Here, $\hat{S}_i$ represents the predicted sensitive attributes, and $n$ is the number of nodes.

### 3.1.2 Contrastive Learning for Informativeness

The model employs contrastive learning to enhance the informativeness of the augmented graphs. This method focuses on ensuring that the node representations between the original graph $h_i$ and the augmented graph $h_i'$ maintain a high degree of similarity, thereby preserving key information. The positive pair in this context is defined as any pair $(h_i, h_i')$, where $h_i$ is the node representation in the original graph and $h_i'$ is the corresponding representation in the augmented graph.

The contrastive learning function $l$, used to compute the loss for these positive pairs, is specified as follows:

$$l(h_i, h_i') = -\log \left( \frac{\exp(\text{sim}(h_i, h_i')/\tau)}{\sum_{j=1}^{n} \exp(\text{sim}(h_i, h_j)/\tau) + \sum_{j=1}^{n} \mathbb{I}_{j \neq i} \exp(\text{sim}(h_i, h_j')/\tau)} \right) \tag{2}$$

where $\tau$ is the temperature scaling parameter, and $\mathbb{I}_{j \neq i}$ is an indicator function that equals 1 if $j \neq i$ and 0 otherwise. This loss function aims to minimize the distance between similar node pairs while maximizing the distance between dissimilar pairs in the embedding space.

The overall contrastive loss $L_{\text{con}}$ is given by:

$$L_{\text{con}} = \frac{1}{2n} \sum_{i=1}^{n} [l(h_i, h_i') + l(h_i', h_i)] \tag{3}$$

### 3.1.3 Reconstruction Based Regularization to Ensure Graph Consistency

To ensure that the augmentation model produces graphs that do not deviate significantly from the input graphs, the model includes a reconstruction-based regularization term in its overall training objective. Specifically, let $L_{BCE}$ and $L_{MSE}$ represent the binary cross-entropy and mean squared error losses, respectively. The regularization term is mathematically expressed as:

$$\begin{aligned} L_{\text{reconst}} &= L_{BCE}(A, \widetilde{A'}) + \lambda L_{MSE}(X, X') \\ &= -\sum_{i=1}^{n} \sum_{j=1}^{n} [A_{ij} \log(\widetilde{A'}_{ij}) + (1 - A_{ij}) \log(1 - \widetilde{A'}_{ij})] + \lambda ||X - X'||_F^2 \end{aligned} \tag{4}$$

Here, $\lambda$ is a hyperparameter, and $\|\cdot\|_F$ denotes the Frobenius norm of a matrix.

### 3.1.4 Integrated Training Objective

By emphasizing informativeness and fairness properties, *Graphair* seeks to produce augmented graphs that are less susceptible to bias. This approach contributes to fairer graph representation learning while preserving the valuable information contained in the data. The overall training process is described as the following min-max optimization procedure, where $\alpha$, $\beta$, and $\gamma$ are hyperparameters that balance the different loss components:

$$\min_{f,g} \max_k L = \min_{f,g} \max_k \alpha L_{\text{adv}} + \beta L_{\text{con}} + \gamma L_{\text{reconst}} \tag{5}$$

## 3.2 Datasets

To replicate the main claims, we use the same datasets as the original paper by Ling et al. (2022), which include specific dataset splits and sensitive and target attributes. We employ three real-world graph datasets: NBA[1], containing player statistics, and two subsets of the Pokec social network from Slovakia, namely Pokec-n and Pokec-z (Dai & Wang, 2021). The specifics of these datasets are summarized in Table 1.

For the link prediction task, we utilize well-established benchmark datasets in this domain: Cora, Citeseer, and Pubmed (Spinelli et al., 2021; Chen et al., 2022; Current et al., 2022; Li et al., 2021). These datasets feature scientific publications as nodes with bag-of-words vectors of abstracts as node features. Edges represent citation links between publications. Notably, these datasets possess a broader range of features compared to those used by Ling et al. (2022) and have a larger set of possible sensitive attributes $|S|$. Full details are provided in Table 1.

Table 1: Dataset Statistics.

| Dataset | $S$ | $|S|$ | Features | Nodes | Edges |
|---------|------|-------|----------|--------|--------|
| NBA | nationality | 2 | 39 | 403 | 16,570 |
| Pokec-z | region | 2 | 59 | 67,797 | 882,765 |
| Pokec-n | region | 2 | 59 | 66,569 | 729,129 |
| Citeseer | paper class | 6 | 3,703 | 3,327 | 9,104 |
| Cora | paper class | 7 | 1,433 | 2,708 | 10,556 |
| PubMed | paper class | 3 | 500 | 19,717 | 88,648 |

## 3.3 Experimental Setup

We obtain the model's codebase from the DIG library, more specifically, from the FairGraph module[2]. To enhance reproducibility, we employ complete seeding across all operations, which was missing in some operations of the original code. A key difference in the experimental setup between the one reported by the original authors and ours is that we conducted a 10,000-epoch grid search for the Pokec dataset, instead of the 500-epoch grid search initially reported by (Ling et al., 2022). This modification was recommended by the original authors to enhance reproducibility. We refer to the subsection A.4 for more details, where we show that a 500-epoch search does not yield optimal results for the Pokec datasets, but higher epochs improve performance in accuracy and fairness. To verify Claims 1 and 2, we follow the procedure described by Ling et al. (2022). For Claim 3, due to memory constraints, we compute the homophily and Spearman correlation values for the Pokec datasets on a mini-batch instead of the entire graphs. Aside from this, we adhere to the same procedures for this claim as well.

## 3.4 Link Prediction

In our study, we extend the scope of the original work by performing a different downstream task, namely link prediction. We implement several modifications to adjust the Graphair network for the downstream task of link prediction. First of all, we modify the output size of the adversarial network from two, which corresponded to the binary sensitive feature in the Pokec and NBA datasets, to the respective number of distinct values of the sensitive feature in the corresponding dataset.

To create the link embeddings for the input of the classifier, we compute the Hadamard product of the node embeddings (Horn & Johnson, 2012), which pairs node embeddings effectively. For nodes $v$ and $u$, we define the link embedding $h_{vu}$ as:

$$h_{vu} = h_v \circ h_u \tag{6}$$

These new link embeddings require labels that reflect the sensitive attributes of the connected nodes. To this end, we integrate dyadic-level fairness criteria into our fairness assessment for link prediction. We form

---

[1]https://www.kaggle.com/datasets/noahgift/social-power-nba
[2]https://github.com/divelab/DIG/tree/dig-stable/dig/fairgraph

dyadic groups that relate sensitive node attributes to link attributes, following the mixed and subgroup dyadic-level fairness principles suggested by Masrour et al. (2020). The mixed dyadic-level groups classify links as either inter- or intra-group based on whether they connect nodes from the same or different sensitive groups. The subgroup dyadic-level approach assigns each link to a subgroup based on the sensitive groups of the nodes it connects. A subgroup is formed for each possible combination of sensitive groups. This method facilitates the measurement of fairness in two distinct aspects. Firstly, it assesses how well each protected subgroup is represented in link formation at the subgroup dyadic-level. Secondly, it evaluates node homogeneity for links at the mixed dyadic-level.

For assessing fairness in link prediction, we aim to optimize for Equality of Opportunity (EO) and Demographic Parity (DP) using these dyadic groups. Following the original paper's definition, DP is defined as:

$$\left| \mathbb{P}(\hat{Y} = 1 \mid D = 0) - \mathbb{P}(\hat{Y} = 1 \mid D = 1) \right|$$

and EO as:

$$\left| \mathbb{P}(\hat{Y} = 1 \mid D = 0, Y = 1) - \mathbb{P}(\hat{Y} = 1 \mid D = 1, Y = 1) \right|$$

where $Y$ is a ground-truth label, $\hat{Y}$ is a prediction, and in the case of link prediction, $D$ denotes the dyadic group to which the link belongs. These definitions extend to multiple dyadic groups ($|D| > 2$), which is the case for subgroup dyadic analysis. As our final metrics, we define the Demographic Parity difference ($\Delta$DP) as the selection rate gap and the Equal Opportunity difference ($\Delta$EO) as the largest discrepancy in true positive rates (TPR) and false positive rates (FPR) across groups identified by $D$:

$$\Delta\text{DP} = \max_d \mathbb{P}(\hat{Y}|D = d) - \min_d \mathbb{P}(\hat{Y}|D = d),$$
$$\Delta\text{TPR} = \max_d \mathbb{P}(\hat{Y} = 1|D = d, Y = 1) - \min_d \mathbb{P}(\hat{Y} = 1|D = d, Y = 1),$$
$$\Delta\text{FPR} = \max_d \mathbb{P}(\hat{Y} = 1|D = d, Y = 0) - \min_d \mathbb{P}(\hat{Y} = 1|D = d, Y = 0),$$
$$\Delta\text{EO} = \max(\Delta\text{TPR}, \Delta\text{FPR}).$$

Link prediction performance is measured using accuracy and the Area Under the ROC Curve (AUC) metrics, representing the trade off between true and false positives.

### 3.4.1 Baselines

We adopt FairAdj (Li et al., 2021) and FairDrop (Spinelli et al., 2021) as our benchmarks. FairAdj learns a fair adjacency matrix during an end-to-end link prediction task. It utilizes a graph variational autoencoder and employs two distinct optimization processes: one for learning a fair version of the adjacency matrix and the other for link prediction. We used two versions of FairAdj: one with 5 epochs ($r = 5$) and the other with 20 epochs ($r = 20$). FairDrop, on the other hand, proposes a biased edge dropout algorithm to counteract homophily and improve fairness in graph representation learning. We used two different versions of FairDrop: one with Graph Convolutional Network (GCN) (Kipf & Welling, 2016) and the other with Graph Attention Networks (GAT) (Velickovic et al., 2017).

### 3.5 Hyperparameters

**Node Classification** To align our experiments closely with the original study, we adopt the hyperparameters specified by the authors. This includes conducting a grid search on the hyperparameters $\alpha$, $\gamma$, and $\lambda$ with the values $\{0.1, 1.0, 10.0\}$, as performed in the original work (Ling et al., 2022). We use the default settings from the original code where specific hyperparameters are not disclosed, a choice validated by the original authors. A complete list of all hyperparameters is provided in Table 7 in subsection A.2.

**Link Prediction** We replicate the grid search from the node classification experiments for link prediction on the Citeseer, Cora, and PubMed datasets. Initially, we conduct a grid search on the model parameters,

including varying the number of epochs, the learning rates for both the Graphair module and the classifier, and the sizes of the hidden layers for both components. We select the most notable model setup based on performance metrics (accuracy and ROC) and fairness values, and then perform a subsequent grid search on the loss hyperparameters $\alpha$, $\lambda$, and $\gamma$ to fine-tune the model further.

We compare the results of Graphair with baseline results from Spinelli et al. (2021) and Li et al. (2021), which also underwent grid searches. For FairAdj, we conducted a grid search focusing on model parameters, involving variations in learning rates, hidden layer sizes, number of outer epochs, and the specific configuration parameter. We evaluated two versions of FairAdj: one with 5 epochs and the other with 20 epochs. For FairDrop, we also performed a grid search on the model parameters, testing different learning rates, epoch counts, and hidden layer sizes. We evaluated two configurations of FairDrop: one using a Graph Convolutional Network (GCN) and the other using Graph Attention Networks (GAT). More detailed information on the hyperparameters fine-tuned during the grid search for each model is presented in Table 8 in subsection A.2.

### 3.6 Computational requirements

All of our experiments are conducted on a high-performance computing (HPC) cluster, that features NVIDIA A100 GPUs, divided into four partitions with a combined memory of 40 GB. For a detailed overview of the GPU hours required for each experiment, see Table 9 in subsection A.3. A rough estimate suggests that a total of 80 GPU hours are necessary to complete all experiments.

## 4 Results

This section presents the outcomes of our experimental results aimed at reproducing and extending the findings of the original work on Graphair. We discuss the reproducibility of specific claims made in the original paper in subsection 4.1 and explore the performance of Graphair in a link prediction downstream task in subsection 4.2.

### 4.1 Results reproducing original paper

**Claim 1:** To verify Claim 1, we performed node classification on the NBA, Pokec-n, and Pokec-z datasets. Table 2 shows a comparison of the results reported by the original authors and the results of baseline models given by the original authors with those obtained by us through replicating the experimental setup described in subsection 3.3. Consistent with the original study, our results are derived by selecting the best outcome from the grid search procedure. We observe that the results for the NBA dataset are similar to those reported by the authors. However, for the Pokec datasets, our Graphair model gets better fairness scores at the cost of worse accuracies.

When examining the fairness-accuracy trade-off in Figure 1, which uses the $\Delta DP$ fairness metric, we see that for the NBA dataset we can achieve a similar trade-off. For the Pokec-z data, a small discrepancy is reflected by a similar trend, but with lower accuracy scores. The Pokec-n dataset also shows a similar trend but fails to reach the higher accuracies of the original model. Considering that the code we used from the DIG library differs from what the original authors used, combined with the fact that a different number of epochs, namely 10,000 was used for the Pokec dataset instead of the originally reported 500, we think there might still be some differences in the experimental setups. Even though these discrepancies are probably minor, they do not allow us to achieve better performance in terms of the accuracy-fairness trade-off for all datasets compared to baseline models, which makes us only partially able to reproduce Claim 1.

Table 2: Comparison of the results reported by the original authors with those obtained by us.

| Methods | NBA | | | Pokec-n | | | Pokec-z | | |
|---|---|---|---|---|---|---|---|---|---|
| | ACC ↑ | $\Delta_{\mathrm{DP}}$ ↓ | $\Delta_{\mathrm{EO}}$ ↓ | ACC ↑ | $\Delta_{\mathrm{DP}}$ ↓ | $\Delta_{\mathrm{EO}}$ ↓ | ACC ↑ | $\Delta_{\mathrm{DP}}$ ↓ | $\Delta_{\mathrm{EO}}$ ↓ |
| FairWalk | $64.54 \pm 2.35$ | $3.67 \pm 1.28$ | $9.12 \pm 7.06$ | $67.07 \pm 0.24$ | $7.12 \pm 0.74$ | $8.24 \pm 0.75$ | $65.23 \pm 0.78$ | $4.45 \pm 1.25$ | $4.59 \pm 0.86$ |
| FairWalk+X | $69.74 \pm 1.71$ | $14.61 \pm 4.98$ | $12.01 \pm 5.38$ | $69.01 \pm 0.38$ | $7.59 \pm 0.96$ | $9.69 \pm 0.09$ | $67.65 \pm 0.60$ | $4.46 \pm 0.38$ | $6.11 \pm 0.54$ |
| GRACE | $70.14 \pm 1.40$ | $7.49 \pm 3.78$ | $7.67 \pm 3.78$ | $68.25 \pm 0.99$ | $6.41 \pm 0.71$ | $7.38 \pm 0.84$ | $67.81 \pm 0.41$ | $10.77 \pm 0.68$ | $10.69 \pm 0.69$ |
| GCA | $\mathbf{70.43 \pm 1.19}$ | $18.08 \pm 4.80$ | $20.04 \pm 4.34$ | $\mathbf{69.34 \pm 0.20}$ | $6.07 \pm 0.96$ | $7.39 \pm 0.82$ | $67.07 \pm 0.14$ | $7.90 \pm 1.10$ | $8.05 \pm 1.07$ |
| FairDrop | $69.01 \pm 1.11$ | $3.66 \pm 2.32$ | $7.61 \pm 2.21$ | $67.78 \pm 0.60$ | $5.77 \pm 1.83$ | $5.48 \pm 1.32$ | $67.32 \pm 0.61$ | $4.05 \pm 1.05$ | $3.77 \pm 1.00$ |
| NIFTY | $69.93 \pm 0.09$ | $3.31 \pm 1.52$ | $4.70 \pm 1.04$ | $67.15 \pm 0.43$ | $4.40 \pm 0.99$ | $3.75 \pm 1.04$ | $65.52 \pm 0.31$ | $6.51 \pm 0.51$ | $5.14 \pm 0.68$ |
| FairAug | $66.38 \pm 0.85$ | $4.99 \pm 1.02$ | $6.21 \pm 1.95$ | $69.17 \pm 0.18$ | $5.28 \pm 0.49$ | $6.77 \pm 0.45$ | $\mathbf{68.61 \pm 0.19}$ | $5.10 \pm 0.69$ | $5.22 \pm 0.84$ |
| Graphair | $69.36 \pm 0.45$ | $2.56 \pm 0.41$ | $\mathbf{4.64 \pm 0.17}$ | $67.43 \pm 0.25$ | $2.02 \pm 0.40$ | $1.62 \pm 0.47$ | $68.17 \pm 0.08$ | $2.10 \pm 0.17$ | $2.76 \pm 0.19$ |
| Graphair (ours) | $68.54 \pm 0.32$ | $\mathbf{1.31 \pm 0.19}$ | $5.34 \pm 0.32$ | $65.76 \pm 0.01$ | $\mathbf{0.72 \pm 0.34}$ | $\mathbf{0.41 \pm 0.40}$ | $65.22 \pm 0.01$ | $\mathbf{1.32 \pm 0.29}$ | $\mathbf{2.24 \pm 0.31}$ |

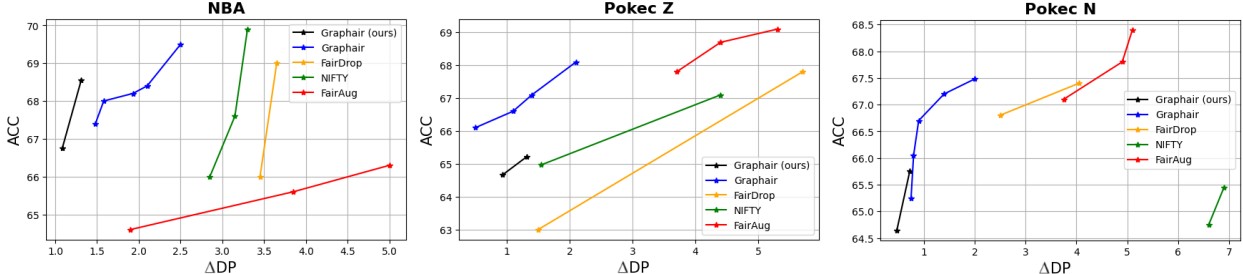

Figure 1: ACC and DP trade-off for baselines, Graphair and our results for Graphair. Upper-left corner (high accuracy, low demographic parity) is preferable.

**Claim 2:** Table 3 confirms that Claim 2 is substantiated for both the NBA and Pokec-z datasets. In comparisons of Graphair with and without feature masking (FM) or edge perturbation (EP), notable increases in fairness metrics are observed. This supports the claim that each model component contributes to mitigating prediction bias. Interestingly, we notice an increase in accuracy when removing feature masking across all datasets, a result that deviates from findings in the original work, which showcased similar accuracy scores when EP was removed. We attribute this to the use of more training epochs in our experimental setup. It seems logical that performance would improve when edge perturbation is removed, as the encoder model $g_{\mathrm{enc}}$ can utilize the original adjacency matrix. This allows the classifier to exploit increased homophily in the network, thereby increasing accuracy and worsening fairness.

Table 3: Comparisons among different components in the augmentation model.

| Methods | NBA | | | Pokec-n | | | Pokec-z | | |
|---|---|---|---|---|---|---|---|---|---|
| | ACC ↑ | $\Delta_{\mathrm{DP}}$ ↓ | $\Delta_{\mathrm{EO}}$ ↓ | ACC ↑ | $\Delta_{\mathrm{DP}}$ ↓ | $\Delta_{\mathrm{EO}}$ ↓ | ACC ↑ | $\Delta_{\mathrm{DP}}$ ↓ | $\Delta_{\mathrm{EO}}$ ↓ |
| Graphair (ours) | $68.54 \pm 0.40$ | $\mathbf{1.31 \pm 0.27}$ | $\mathbf{5.34 \pm 0.24}$ | $65.76 \pm 0.02$ | $\mathbf{0.72 \pm 0.36}$ | $\mathbf{0.41 \pm 0.42}$ | $65.22 \pm 0.02$ | $\mathbf{1.32 \pm 0.33}$ | $\mathbf{2.24 \pm 0.35}$ |
| Graphair w/o EP (ours) | $\mathbf{72.68 \pm 0.40}$ | $2.95 \pm 1.12$ | $9.05 \pm 2.53$ | $\mathbf{67.26 \pm 0.16}$ | $2.11 \pm 0.09$ | $1.37 \pm 0.33$ | $\mathbf{69.25 \pm 0.10}$ | $6.56 \pm 0.74$ | $6.73 \pm 0.65$ |
| Graphair w/o FM (ours) | $67.79 \pm 0.32$ | $10.73 \pm 1.12$ | $26.79 \pm 2.53$ | $64.61 \pm 0.36$ | $3.83 \pm 0.47$ | $3.18 \pm 0.27$ | $57.91 \pm 0.13$ | $5.19 \pm 0.74$ | $6.34 \pm 0.33$ |

**Claim 3:** The plots in Figure 2 clearly illustrate that the homophily values for the augmented graph are lower than those for the original graph. These results support the authors' claim that Graphair automatically generates new graphs with a fairer node topology. The plots in Figure 3 reveal that, for all three datasets, the features with the highest Spearman correlation to the sensitive feature generally exhibit lower values in the fair view. These findings lend support to the authors' claim that Graphair produces features that are fairer. We can therefore conclude that Claim 3 is fully reproducible.

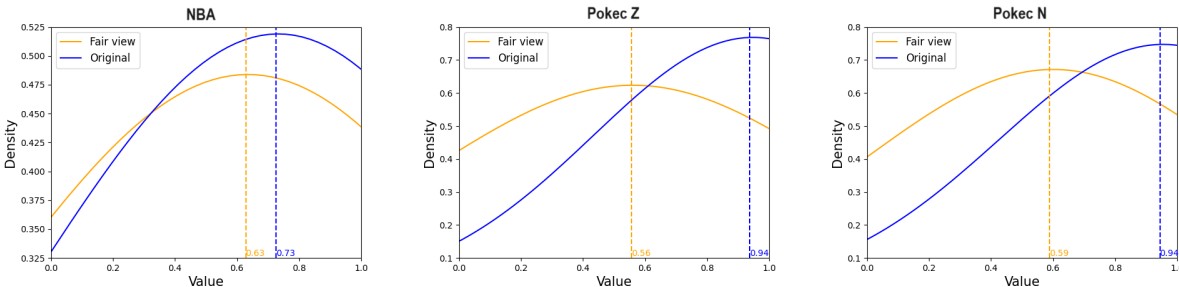

Figure 2: Node sensitive homophily distributions in the original and the fair graph data.

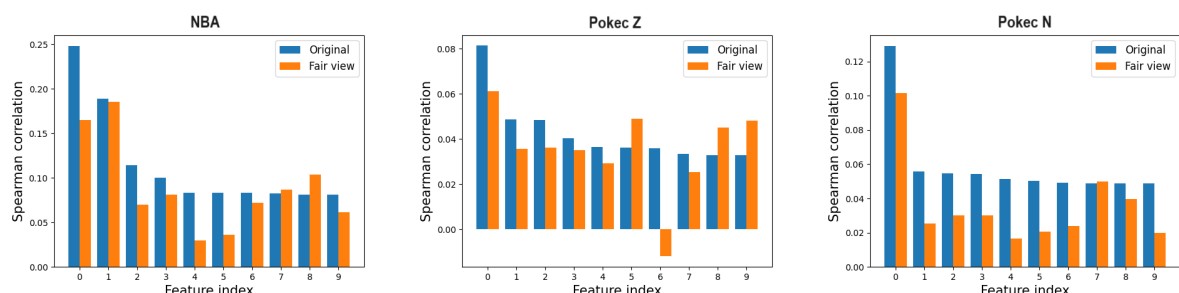

Figure 3: Spearman correlation between node features and the sensitive attribute in the original and the fair graph data.

## 4.2 Results beyond original paper

The performance of Graphair compared to baseline models is presented in Table 4, Table 5, and Table 6. For fairness metrics, subscripts $m$ and $s$ denote application to mixed groups and subgroups, respectively. Our findings indicate that Graphair is outperformed by both FairDrop variants in terms of accuracy and AUC across all datasets. However, compared to FairDrop, Graphair excels in all fairness metrics on all datasets, particularly in the subgroup variants on the Citeseer and Cora datasets. FairAdj demonstrates comparable performance in accuracy and AUC relative to Graphair but performs worse in subgroup fairness metrics, only excelling in the $\Delta DP_m$ metric across all datasets.

We further investigate the trade-off between accuracy and $\Delta DP$ for both mixed and subgroup variants across the three datasets for each model, as shown in Figure 4. The trade-off for Graphair is comparable to the baseline models for the mixed variant, while achieving a superior trade-off for the subgroup variant. While Spinelli et al. (2021) suggest that high scores for subgroup fairness metrics are due to dataset characteristics, we find that Graphair improves this metric through its augmentative approach, which generates augmentations with similar predictive power across different subgroups. This enables Graphair to consistently predict links with the same probability across various subgroups, resulting in lower subgroup dyadic-level fairness while maintaining predictive power.

Table 4: Link Prediction on Citeseer

| Method | Accuracy ↑ | AUC ↑ | $\Delta DP_m \downarrow$ | $\Delta EO_m \downarrow$ | $\Delta DP_s \downarrow$ | $\Delta EO_s \downarrow$ |
|---|---|---|---|---|---|---|
| FairAdj$_{r=5}$ | $77.1 \pm 2.5$ | $83.4 \pm 2.3$ | $33.8 \pm 3.5$ | $8.2 \pm 4.1$ | $65.6 \pm 6.2$ | $80.0 \pm 9.0$ |
| FairAdj$_{r=20}$ | $73.5 \pm 2.8$ | $81.2 \pm 3.0$ | $\mathbf{26.0 \pm 3.4}$ | $5.3 \pm 3.5$ | $56.5 \pm 7.5$ | $70.1 \pm 8.0$ |
| GCN+FairDrop | $\mathbf{87.3 \pm 1.7}$ | $\mathbf{97.1 \pm 1.6}$ | $46.7 \pm 3.0$ | $8.8 \pm 4.5$ | $68.9 \pm 6.0$ | $41.1 \pm 10.0$ |
| GAT+FairDrop | $86.3 \pm 1.3$ | $96.6 \pm 1.2$ | $45.0 \pm 2.7$ | $9.6 \pm 4.8$ | $68.4 \pm 5.9$ | $39.3 \pm 9.8$ |
| Graphair (ours) | $79.2 \pm 0.7$ | $86.9 \pm 0.6$ | $40.5 \pm 0.6$ | $\mathbf{1.1 \pm 1.5}$ | $\mathbf{2.9 \pm 3.0}$ | $\mathbf{11.3 \pm 3.8}$ |

Table 5: Link Prediction on Cora

| Method | Accuracy ↑ | AUC ↑ | $\Delta\mathrm{DP}_m \downarrow$ | $\Delta\mathrm{EO}_m \downarrow$ | $\Delta\mathrm{DP}_s \downarrow$ | $\Delta\mathrm{EO}_s \downarrow$ |
|---|---|---|---|---|---|---|
| FairAdj$_{r=5}$ | $76.6 \pm 1.9$ | $83.8 \pm 2.4$ | $38.9 \pm 4.5$ | $11.6 \pm 4.7$ | $78.5 \pm 5.3$ | $100.0 \pm 8.0$ |
| FairAdj$_{r=20}$ | $73.0 \pm 1.8$ | $78.9 \pm 2.2$ | $\mathbf{29.3 \pm 3.1}$ | $\mathbf{4.7 \pm 4.7}$ | $83.3 \pm 7.2$ | $97.5 \pm 7.8$ |
| GCN+FairDrop | $\mathbf{90.4 \pm 1.1}$ | $\mathbf{97.0 \pm 0.8}$ | $58.2 \pm 2.7$ | $17.3 \pm 5.2$ | $93.6 \pm 3.7$ | $98.5 \pm 0.5$ |
| GAT+FairDrop | $85.4 \pm 1.4$ | $96.2 \pm 1.2$ | $53.2 \pm 3.0$ | $17.8 \pm 4.6$ | $84.7 \pm 2.3$ | $98.2 \pm 0.5$ |
| Graphair (ours) | $75.2 \pm 0.8$ | $83.3 \pm 0.9$ | $38.7 \pm 0.8$ | $12.7 \pm 1.7$ | $\mathbf{37.9 \pm 3.2}$ | $\mathbf{13.3 \pm 2.1}$ |

Table 6: Link Prediction on PubMed

| Method | Accuracy ↑ | AUC ↑ | $\Delta\mathrm{DP}_m \downarrow$ | $\Delta\mathrm{EO}_m \downarrow$ | $\Delta\mathrm{DP}_s \downarrow$ | $\Delta\mathrm{EO}_s \downarrow$ |
|---|---|---|---|---|---|---|
| FairAdj$_{r=5}$ | $83.7 \pm 3.2$ | $90.6 \pm 4.7$ | $40.76 \pm 3.7$ | $35.5 \pm 2.5$ | $40.7 \pm 3.2$ | $16.7 \pm 1.9$ |
| FairAdj$_{r=20}$ | $76.1 \pm 1.8$ | $83.7 \pm 2.1$ | $\mathbf{37.2 \pm 2.5}$ | $22.6 \pm 2.0$ | $53.4 \pm 5.5$ | $39.9 \pm 5.4$ |
| GCN+FairDrop | $\mathbf{92.3 \pm 0.4}$ | $\mathbf{97.4 \pm 0.2}$ | $44.2 \pm 0.5$ | $7.6 \pm 0.7$ | $54.1 \pm 1.5$ | $15.3 \pm 2.6$ |
| GAT+FairDrop | $92.2 \pm 0.8$ | $97.3 \pm 0.7$ | $43.7 \pm 0.9$ | $7.8 \pm 1.1$ | $54.5 \pm 2.1$ | $14.3 \pm 4.1$ |
| Graphair (ours) | $82.3 \pm 0.2$ | $89.8 \pm 2.9$ | $37.6 \pm 0.4$ | $\mathbf{2.6 \pm 0.7}$ | $\mathbf{36.4 \pm 2.4}$ | $\mathbf{8.0 \pm 2.4}$ |

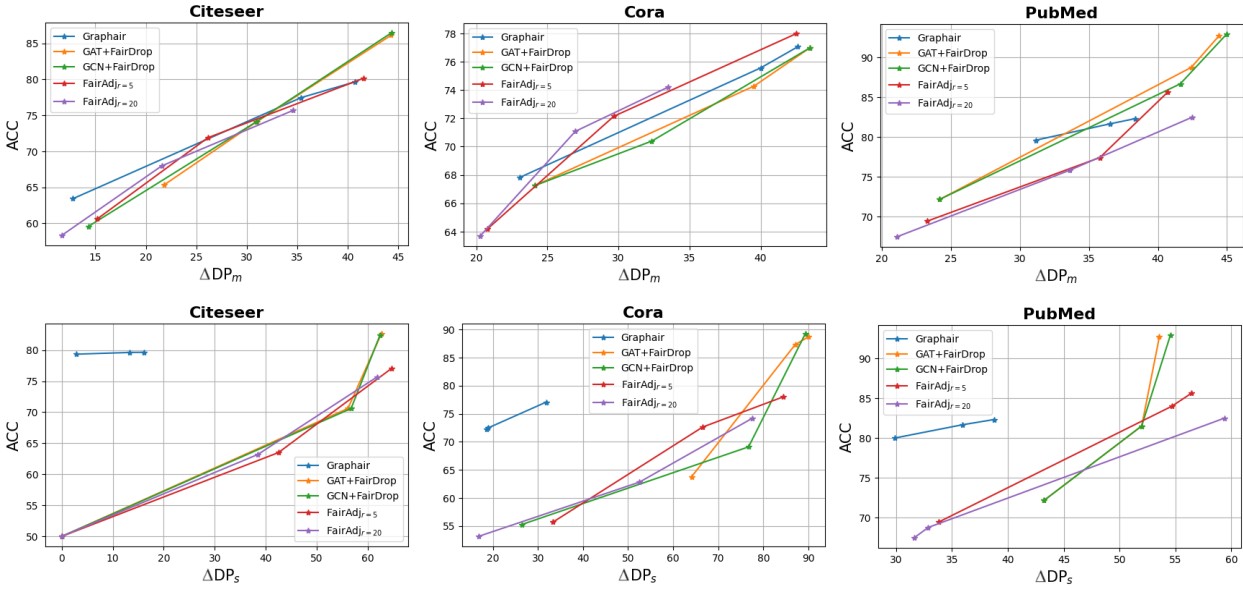

Figure 4: ACC and DP trade-off for the baselines and our Graphair for link prediction. The top row shows the $\Delta\mathrm{DP}_m$ metric, and the bottom row shows the $\Delta\mathrm{DP}_s$ metric. Points in the upper-left corner are desired.

## 5  Discussion

Upon revisiting the three claims in our study, we find that Claim 1 is partially reproducible, whereas Claims 2 and 3 are fully reproducible. In the case of Claim 1, while we were able to replicate the performance of the NBA dataset consistent with the original paper, discrepancies emerged with the Pokec datasets. Specifically, our results showed improved fairness scores at the expense of lower accuracies compared to the original findings. This could be attributed to differences in experimental setup, particularly the number of training epochs used, which deviated from the original study's methods. We used 10,000 epochs for the Pokec datasets as opposed to the 500 reported in the original paper, a change recommended by the original authors.

Further analysis of Graphair's performance in link prediction indicates that, while it demonstrates a comparable fairness-accuracy trade-off to baseline models for mixed dyadic-level fairness, Graphair has a superior trade-off for subgroup dyadic-level fairness.

### 5.1 What was easy and what was difficult

The clarity of the code within the DIG library[3] significantly facilitated reproducibility. The original paper provided a clear outline of the experiments, enabling a straightforward process to identify the necessary components for reproducing the study's claims and implementing our link prediction extensions.

We encountered initial challenges with the reproducibility of Claim 1, which necessitated seeking clarification from the authors. Correspondence with the original authors resolved issues related to unspecified hyperparameter settings and a bug in the code. Reproducing Claim 3 for the Pokec datasets proved non-trivial due to the large memory requirements for processing the full graph, necessitating solutions to acquire experimental results.

### 5.2 Communication with original authors

We initiated contact with one of the authors, Hongyi Ling, via email to seek clarification on our initial results that did not match those of the original paper. These discrepancies were resolved, and the authors responded promptly to our emails, providing valuable feedback. Most notably, they recommended a change in our experimental setup for the Pokec datasets, specifically increasing the number of epochs from 500 to 10,000.

---

[3]https://github.com/divelab/DIG

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

# A   Appendix

## A.1   Overview of the Graphair model

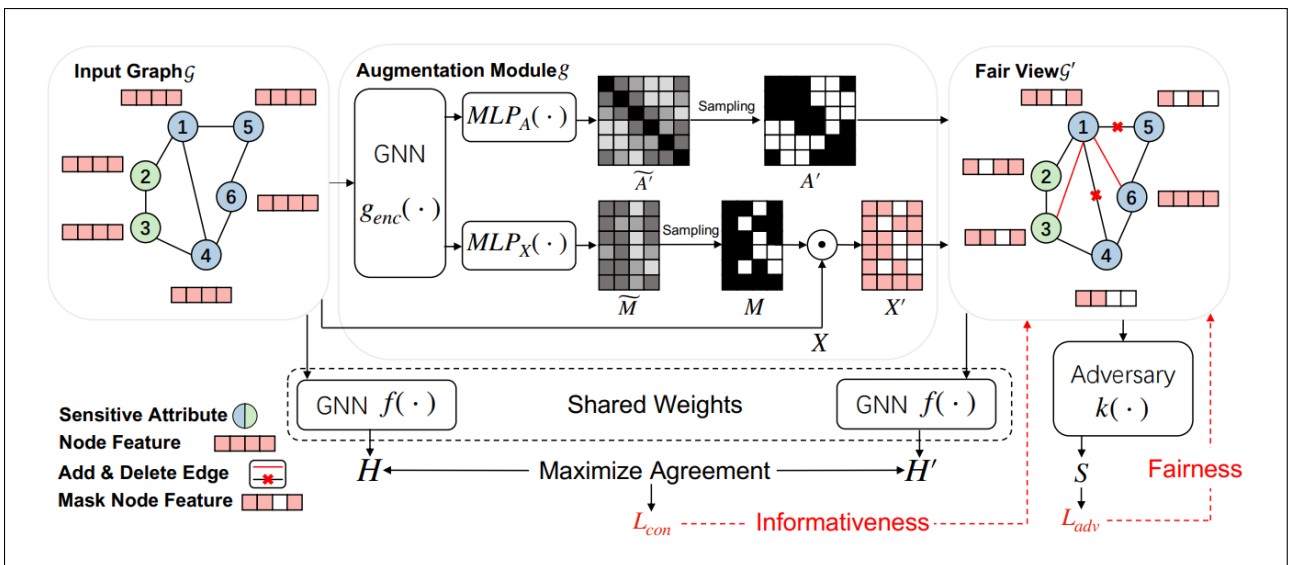

Figure 5: Overview of the Graphair framework (Ling et al., 2022)

## A.2   List of all hyperparameters

Table 7: Overview of hyperparameters for the Graphair model $m$ and the evaluation classifier $c$ on all datasets.

| Hyperparameter | NBA | Pokec-n | Pokec-z | Citeseer | Cora | PubMed |
|---|---|---|---|---|---|---|
| $\alpha$ | 1.0 | 0.1 | 10.0 | 0.1 | 10.0 | 10.0 |
| $\beta$ | 0.1 | 1.0 | 10.0 | 0.1 | 10.0 | 10.0 |
| $\gamma$ | 0.1 | 0.1 | 0.1 | 0.1 | 0.1 | 0.1 |
| $\lambda$ | 1.0 | 10.0 | 10.0 | 1.0 | 10.0 | 0.1 |
| $c_{\text{hidden}}$ | 128 | 128 | 128 | 128 | 128 | 128 |
| $c_{\text{learning\_rate}}$ | 1e-3 | 1e-3 | 1e-3 | 5e-3 | 5e-3 | 5e-3 |
| $c_{\text{weight\_decay}}$ | 1e-5 | 1e-5 | 1e-5 | 1e-5 | 1e-5 | 1e-5 |
| $m_{\text{learning\_rate}}$ | 1e-4 | 1e-4 | 1e-4 | 1e-4 | 1e-4 | 1e-4 |
| $m_{\text{weight\_decay}}$ | 1e-5 | 1e-5 | 1e-5 | 1e-5 | 1e-5 | 1e-5 |

Table 8 presents a comprehensive overview of the hyperparameters adjusted during the grid search. The initial seven rows correspond to the Graphair model, while the subsequent rows correspond to the baseline models used for link prediction tasks. Regarding the epoch count for node classification with Graphair, 500 epochs were used for the NBA dataset and 10,000 for the Pokec datasets. For grid searches with the Graphair model for link prediction, the number of epochs was set to 200.

Table 8: Overview of all hyperparameters tuned in the grid search.

| Hyperparameter | Node Classification | Link Prediction |
|---|---|---|
| $\alpha$ | { 0.1, 1, 10} | { 0.1, 1, 10} |
| $\gamma$ | { 0.1, 1, 10} | { 0.1, 1, 10} |
| $\lambda$ | { 0.1, 1, 10} | { 0.1, 1, 10} |
| Classifier lr | 1e-3 | {1e-2, 1e-3, 1e-4} |
| Model lr | 1e-4 | {1e-2, 1e-3, 1e-4} |
| Classifier Hidden Dimension | 128 | {64, 128, 256} |
| Model Hidden Dimension | 128 | {64, 128, 256} |
| Model lr (FairAdj)* | - | {0.1, 1e-2, 1e-3} |
| hidden1 (FairAdj) | - | {16, 32, 64} |
| hidden2 (FairAdj) | - | {16, 32, 64} |
| outer epochs (FairAdj) | - | {4, 10, 20} |
| Epochs(FairDrop) | - | {100, 200, 500, 1000} |
| Model lr (FairDrop) | - | {5e-2, 1e-2, 5e-3, 1e-3, 5e-4, 1e-4} |
| Hidden dim. (FairDrop) | - | {64, 128, 256, 512} |

### A.3 GPU Run hours

Table 9: Computational Requirement Overview

| Name of the experiment | GPU Hour (Hour) | Max GPU Memory Usage (GB) |
|---|---|---|
| Claim 1 (grid search) | 20 | 3.70 |
| Claim 2 | 2 | 3.67 |
| Claim 3 | 0.5 | 3.70 |
| Link Prediction (grid search) | 60 | 3.70 |

GPU hour is measured by the amount of time each experiment script needed from the start to the end. Maximum GPU memory usage is determined by *max_memory_allocated* method from the Pytorch library.

### A.4 Impact of the number of epochs on the accuracy and fairness results

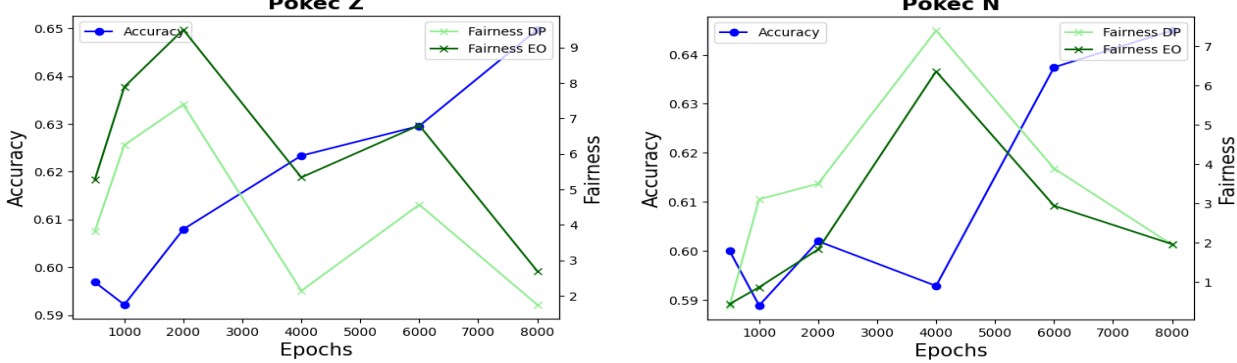

Figure 6: Impact of the number of epochs on the accuracy and fairness results.

Following the original authors' recommendation to increase the number of epochs in order to replicate their results, we performed an ablation study to examine the impact of the number of epochs on accuracy and fairness metrics. As illustrated in Figure 6, there is a positive correlation between the number of epochs and

both accuracy and fairness metrics. Based on these findings, we conducted our experiments using 10,000 epochs for the Pokec datasets as this led to a notable increase in performance.

