# OpenReview forum: "Reproducibility Study Of Learning Fair Graph Representations Via Automated Data Augmentations"
_TMLR — Accepted by TMLR_

### Review · Reviewer_yRgK · 2024-03-11

**Summary Of Contributions:**

This paper studies the reproducibility of the paper "Learning fair graph representations via automated data augmentations" as well as extending the scope of the paper (method of which is called GraphAir) to link prediction. The reproduction experiments of GraphAir shows that two of the three claims are partially verified while the third one is fully verified. In addition, the authors experiment GraphAir on link prediction tasks, with results showing the good performances of GraphAir.

**Audience:**

Yes

**Broader Impact Concerns:**

No.

**Claims And Evidence:**

No

**Requested Changes:**

1. Please show the training stability issue of GraphAir, and discuss whether some common ways to mitigate instability (e.g. gradient clipping, regularization, larger batches, etc.) will work to mitigate it.
2. Please provide some insights about why the paper classes are used as sensitive attributes. I have a slight concern that this attribute may not act similarly as commonly considered sensitive attributes (e.g. gender, age, etc.).
3. Please consider some better evaluation methods to jointly evaluate both the fairness and the accuracy for Tables 4-6.

**Strengths And Weaknesses:**

## Strengths
1. The reproduced paper GraphAir, as of March 2024 has 29 citations so it has reasonable impact. Therefore, a reproduction of it will be of interest to some of the audience of TMLR.
2. The paper is generally well written and easy to follow.

## Weaknesses
Before I state the weaknesses of this paper, I would admit that I am not an expert in fair graph learning, so my comments are from a generally high level.
1. Not sure about the 'training instability of GraphAir'. This paper states in Section 4.1 that 'this issue might also stems from the observed training instability', but no evidence regarding the training instability is given or discussed, which makes it hard to understand the detailed cause of this paper.
2. I am not so sure about the setting of citation datasets. According to the authors, the sensitive attributes are paper classes. I am wondering why the setting is done as such, as paper classes are generally not considered sensitive. Moreover, the paper classes are just the node labels and may be very directly related to the citation relations. Therefore, I am a bit confused about why the setting is done as such. Of course, the authors say that this follows another paper (Spinelli et al. 2021), which is fine. However I do like to know some rationale underlying it.
3. I am not so convinced by the results in Table 4-6. As can be seen in Table 4-6, there is a very clear tradeoff between fairness and accuracy, and GraphAir commonly excels on fairness but not so on Accuracy. I think it is difficult to draw a conclusion about which method is definitely better, or whether the performance gap is large or small, as comparison on two dimensions cannot be solely determined by one dimension. Are there any ways to make the comparison more thorough by involving both dimensions (accuracy, fairness)?

---

> ### Author Response · Authors · 2024-06-15
> **Response to Reviewer yRgK regarding the requested changes**
>
> Thank you for reviewing our paper. We appreciate your constructive feedback and hope to have made the necessary revisions accordingly. Below, we address each of your concerns in detail:
>
> 1. In our initial submission, we noted that training instability was likely due to a bug in the provided code. We communicated this issue with the original authors, who have since resolved it. Consequently, we repeated the experiment to assess if the worse performance for the Pokec datasets presented in Table 2 could be addressed. Our new results show a much smaller discrepancy between the results from the original authors and our grid search results. Additionally, we have included the fairness-accuracy curve for our Graphair model and compared it to the one reported by the original authors. A small discrepancy persists, which we suspect stems from our change in the experimental setup from 500 to 10,000 epochs for the Pokec datasets, as recommended by the original authors; this is discussed in more detail in the revised version. Furthermore, we reran experiments for Claim 2 and are now able to fully substantiate the claim.
> 2. We selected Citeseer, Cora, and PubMed as our link prediction datasets because they are recognized as common benchmark networks in multiple studies within the fairness domain (Spinelli et al., 2021; Chen et al., 2022; Current et al., 2022; Li et al., 2021). Your point regarding the selection of academic article classes/categories as sensitive features is accurate. Indeed, this issue is not addressed in the referenced papers. Nonetheless, we believe that our model's capability to perform well across various sensitive features remains valid, given that sensitive features can vary depending on the specific context. Regardless of the sensitive features we consider, the model should act fairly.
> 3. We have expanded our results on the trade-off between fairness and accuracy, which are now detailed in Figure 4 and discussed in Section 4.2 of our revised paper. The insights gained led us to revise our claims about the performance of Graphair compared to baseline models, and we have adjusted our statements accordingly. We are currently still encountering a memory issue with the PubMed dataset when using the FairAdj model, which restricts us to running experiments with only a limited set of hyperparameters. We are actively working to resolve this issue and will likely need to revise our paper once more to incorporate these results. Despite this, we believe the results obtained thus far are nearly representative of this method's optimal performance. We do not anticipate that new, potentially improved results would necessitate revising the claims we have made regarding the comparison of Graphair to its baseline variants in the link prediction domain. But if that would be the case, we will of course try and revise this accordingly. EDIT: We have identified the issue and uploaded a revised version. This version now concludes our revisions, and the paper is fully ready for re-review.

---

### Review · Reviewer_mrv3 · 2024-05-02

**Summary Of Contributions:**

The authors attempt to reproduce the results of the paper "Learning Fair Graph Representations Via Automated Data Augmentations" by Ling et al. (2022), where the Graphair framework was proposed. Furthermore, the authors extend the Graphair framework from the original node classification setting in Ling et al. (2022) to link prediction. The main findings are as follows:
- One of the claims in Ling et al. (2022) can be fully verified, while the other two can only be partially verified.
- The Graphair framework also performs quite well on the link prediction, resulting in smaller deviations from fairness.

**Audience:**

Yes

**Broader Impact Concerns:**

None that I observe.

**Claims And Evidence:**

No

**Requested Changes:**

Major issues:
- Table 2 only shows a comparison between your implementation of Graphair and the results reported by the authors. The other methods that Ling et al. (2022) compare against are missing, so it is difficult to interpret the results. After examining Table 1 in Ling et al. (2022), I see that the accuracy values you are obtaining on the Pokec data sets are far lower than the other baselines, which is a large difference that needs to be accounted for. To further investigate the discrepancy you observe,  consider expanding the scope of your replication study to include additional methods. Perhaps this will help you to identify what is causing the discrepancy with Graphair.
- Description of results for Claim 2 in Section 4.1 are misleading: "we observe clear increases in both fairness metrics and accuracy". Indeed, fairness metrics are improving, but accuracy is not. This discussion should be revised.

Minor issues:
- What is the FairAdj method in Section 4.2? It is not discussed at all. Add at least a citation and brief description.
- Hyperparameter $\lambda$ in Section 3.3 has not been mentioned previously in the paper, unlike the other 3 hyperparameters.
- Fairness metrics reported in Tables 4 and 5 have $m$ and $s$ subscripts, which are not described. I assume these correspond to the mixed and subgroup fairness principles discussed in Section 3.4.1.

**Strengths And Weaknesses:**

Strengths:
- Not only attempts to replicate prior work, but also extends its scope to link prediction.
- Addresses a topic that should be of high interest to TMLR readers.

Weaknesses:
- Findings are of questionable value to the community due to the limited scope of the study. It is unclear why the results obtained by the authors for Claims 1 and 2 differ so much from Ling et al. (2022). I elaborate further in Requested Changes.
- Several presentation issues. See both the major and minor issues listed in Requested Changes.

---

> ### Author Response · Authors · 2024-06-15
> **Response to Reviewer mrv3 regarding the major and minor issues**
>
> Thank you for reviewing our paper. We appreciate your constructive feedback and hope to have made the necessary revisions accordingly. Below, we address each of your concerns in detail:
>
> Major Issues:
>
> * We have incorporated the baseline methods mentioned by the original authors into Table 2. After the training instability issue was resolved by the original authors, we repeated the experiments to determine if the significantly lower scores presented in Table 2 could be reconciled. Our new results show a much smaller discrepancy between the results from the original authors and our grid search results. Additionally, we have added the fairness accuracy curve of our Graphair model and compared it to the one reported by the original authors. While a small discrepancy remains, we suspect this is due to changing our experimental setup to 10,000 epochs for the Pokec dataset, as recommended by the original authors; we elaborate on this in the revised version.
> * Indeed, the table presented mixed results. We acknowledge the previously unclear wording. The bug fix allowed us to rerun this claim, and we are now able to fully substantiate Claim 2. We have revised this part accordingly.
>
> Minor Issues:
>
> * We have included a brief description of both models in Section 3.4.1.
> * The hyperparameter lambda was indeed not mentioned previously. We have now addressed this by adding a paragraph in Section 3.1.3.
> * Yes, that is correct. We have added a short clarification in Section 4.2
>
> Additionally, we have addressed issues raised by other reviewers concerning the evidence for the statements made about the link prediction task. We have revised our previous claims, conducted more extensive grid searches, and added accuracy and fairness trade-off curves to compare performance.

---

### Review · Reviewer_QVW1 · 2024-05-30

**Summary Of Contributions:**

This paper conducts a reproducibility analysis of Graphair (ICLR 2023) with an extensive grid search, which results in new findings different from the existing papers. The paper also explores wider applications of Graphair.

**Audience:**

Yes

**Claims And Evidence:**

Yes

**Requested Changes:**

Please address my concerns above.

**Strengths And Weaknesses:**

The strengths of this paper are as follows.

- The reproducibility analysis is welcome for TMLR, which expects diverse and high-practicality submissions.
- The authors have established a more reasonable experimental setup compared with the original paper of ICLR 2023.
- The authors have released their codes.

The weaknesses of this paper are as follows.

- The authors have ignored a careful grid search of the other compared baselines. If the proposed method does not use grid search, it is questionable for the compared methods.
- The Graphair has poor accuracy while achieving high accuracy. As shown in Table 4, Table 5, and Table 6, Graphair is far worse on accuracy-based metrics: Accuracy and AUC. As we know, for graph tasks such as link prediction, etc., the drop in 0.01-level on AUC could be a significant drop. Therefore, Graphair is still not so powerful even if the authors have conducted a careful grid search.
- Although reproducibility analysis is a good topic for TMLR, we would like to see more insightful observations in addition to the original paper, since TMLR has very high standards.

---

> ### Author Response · Authors · 2024-06-15
> **Response to Reviewer QVW1 regarding the grid search of baselines and the trade-off between accuracy and fairness.**
>
> Thank you for reviewing our paper. We appreciate your constructive feedback and hope to have made the necessary revisions accordingly. Below, we address each of your concerns in detail:
> * We performed an extensive grid search on the baseline models presented in Section 4.2 of our link prediction study. Additionally, we conducted a more thorough grid search on the Graphair model. Based on the new results, we have updated the tables and revised the claims regarding the performance of Graphair compared to the baseline models.
> * We have included new results on the trade-off between accuracy across three datasets, as depicted in Figure 4 and updated our results in tables 4-6. These new results show that Graphair's performance is quite similar to that of the baseline models in terms of fairness-accuracy; we elaborate further in Section 4.2, where we have revised our claims regarding Graphair’s performance compared to baseline methods. However, we should say that we are currently encountering a memory issue with the PubMed dataset when using the FairAdj model, which restricts us to running experiments with only a limited set of hyperparameters. We are actively working to resolve this issue and will likely need to revise our paper once more to incorporate these results. Despite this, we believe the results obtained thus far are nearly representative of this method's optimal performance. We do not anticipate that new, potentially improved results would necessitate revising the claims we have made regarding the comparison of Graphair to its baseline variants in the link prediction domain. But if that would be the case, we will of course revise this accordingly.
> EDIT: We have identified the issue and uploaded a revised version. This version now concludes our revisions, and the paper is fully ready for re-review.
> * The original authors resolved a bug in their code, which has allowed us to more closely reproduce Claim 1 and now fully reproduce Claim 2 as well. We have added an additional figure that illustrates the fairness-accuracy trade-off and compares our model with that originally reported by the authors. We have also commented on small discrepancies that remain.

---

### Decision · Action_Editor_ouKE · 2024-08-16

**Recommendation:** Accept with minor revision

**Comment:**

Two of the reviewers recommended to accept the paper while one recommended rejection. The points raised by the reviewer recommending rejection were (1) a lack of grid search for the baselines, which have been addressed in the revision and (2) a concern regarding the relevance of reproducibility analysis for TMLR. Overall, and without further details from the reviewer, I believe the paper should be accepted.

When preparing the final version of the paper, please address the following two points raised by reviewer mrv3:

- Section 3.1.3: lambda is missing in front of Frobenius norm in the second line of the equation.
- Inappropriate use of text citations without parentheses in several locations, such as Section 3.4.1: "FairAdj Li et al. (2021) and FairDrop Spinelli et al. (2021)" should both be replaced with the parenthesis-based citations. This also appears in some other locations, such as Section 3.4: "the node embeddings Horn & Johnson (2012)"

**Audience:**

Yes, reporducibility analysis are of interest to TMLR's audience.

**Claims And Evidence:**

This is a reproducibility analysis submission on the paper "Learning Fair Graph Representations Via Automated Data Augmentations" by Ling et al. (2022). The claims made by the authors are overall supported by substantial evidence. One of the reviewers had concerns regarding the use of grid search for the compared baseline but this seems to have been addressed in the revision.